LINC00844 promotes proliferation and migration of hepatocellular carcinoma by regulating NDRG1 expression

Zhou Wei 1
Huang Kang 1
Zhang Qiuyan 1
Ye Shaojun 1
Zhong Zibiao 1
Zeng Cheng 1
Peng Guizhu 1
Li Ling Wb001019@whu.edu.cn 1
Ye Qifa yqf_china@163.com 1 2
1 Zhongnan Hospital of Wuhan University, Institute of Hepatobiliary Diseases of Wuhan University, Transplant Center of Wuhan University, Hubei Key Laboratory of Medical Technology on Transplantation , Wuhan , China
2 The 3rd Xiangya Hospital of Central South University, Research Center of National Health Ministry on Transplantation Medicine Engineering and Technology , Changsha , China
Bartolini Barbara
Electronic publication date: 2020 Jan 28
Publication date: 2020
Volume: 8
Electronic Location ID: e8394
Received 2019 Sep 13; Accepted 2019 Dec 13
Copyright: ©2020 Zhou et al.
Copyright year: 2020
Copyright holder: Zhou et al.
License: This is an open access article distributed under the terms of the Creative Commons Attribution License, which permits unrestricted use, distribution, reproduction and adaptation in any medium and for any purpose provided that it is properly attributed. For attribution, the original author(s), title, publication source (PeerJ) and either DOI or URL of the article must be cited.
License URL: https://creativecommons.org/licenses/by/4.0/

Keywords: Hepatocellular carcinoma, LINC00844, NDRG1, Proliferation, Migration, Invasion

Funding: Open Research Fund Program of the State Key Laboratory of Virology of China 2018KF005 National Natural Science Foundation of China-Xinjiang joint fund U1403222 Health Commission of Hubei Province scientific research project 2019H074 This project was supported by the Open Research Fund Program of the State Key Laboratory of Virology of China (Grant number: 2018KF005), the National Natural Science Foundation of China-Xinjiang joint fund (Grant number: U1403222) and the Health Commission of Hubei Province scientific research project (Grant number: 2019H074). The funders had no role in study design, data collection and analysis, decision to publish, or preparation of the manuscript.

==============================
Background

Aberrant expression of long noncoding RNAs are implicated in the pathogenesis of human malignancies. LINC00844 expression is dramatically downregulated in prostate cancer, and functional studies have revealed the association between the aberrant expression of LINC00844 and prostate cancer cell invasion and metastasis. However, the function and mechanism of action of LINC00844 in the pathogenesis of hepatocellular carcinoma (HCC) are poorly understood.

Methods

LINC00844 and N-Myc downstream-regulated 1 (NDRG1) expression in HCC tissues and cell lines was detected with real-time quantitative polymerase chain reaction (RT-qPCR) and western blot analysis. Correlations between LINC00844 expression level and clinicopathological features were investigated using the original data from The Cancer Genome Atlas (TCGA) database. HepG2 and HCCLM9 cell lines were transfected with Lv-LIN00844 virus to obtain LINC00844-overexpressing cell lines. Cell proliferation and cell invasion and migration were examined with the cell counting kit-8 (CCK-8) and transwell assay, respectively. Furthermore, the correlation between LINC00844 and NDRG1 expression was analysed using Pearson’s correlation analysis.

Results

LINC00844 expression was significantly downregulatedin HCC tissues and cell lines, and a statistical correlation was detected between low LINC00844 expression and sex (Female), advanced American Joint Committee on Cancer (AJCC) stage (III + IV), histological grade (G3 + G4), and vascular invasion (Micro and Macro). In vitro experiments showed that LINC00844 overexpression significantly repressed the proliferation, migration, and invasion of HCC cells. NDRG1 expression was higher in HCC tissues and LINC00844 could partly inhibit the expression of NDRG1.

Introduction

Hepatocellular carcinoma (HCC) is a common primary liver tumour and one of the most common causes of cancer-related deaths worldwide (Bray et al., 2018). Hepatitis B virus (HBV) infection, chronic Hepatitis C virus (HCV) infection, alcohol abuse, aflatoxin B1 intake, and metabolic syndrome are high risk factors associated with liver cancer. The incidence of HBV infection is the highest in Asia (Forner, Reig & Bruix, 2018). There is still no cure for chronic HBV infection (Ji & Hu, 2017). Despite tremendous progress in HCC therapy, an effective strategy to cure HCC is still missing (Forner, Reig & Bruix, 2018). As observed with many other cancers, HCC tumourigenesis and progression is a complex process involving genetic and epigenetic transformations in pivotal growth regulatory genes (Furuta et al., 2010; Forner, Llovet & Bruix, 2012). Therefore, it is important to clearly elucidate the molecular mechanism underlying the occurrence of HCC and develop novel strategies for the early diagnosis and treatment of patients with HCC.

Evidence suggests that long noncoding RNAs (lncRNAs) play an important role in the regulation of several biological processes related to hepatocarcinogenesis, including cell proliferation, apoptosis, metastasis, angiogenesis, and chemosensitivity (Lin et al., 2018; Deng et al., 2015; Zhuang et al., 2019; Zhang et al., 2019). Many lncRNAs play crucial roles in the oncogenesis and development of malignant tumours and function as oncogenes or tumour suppressors (Bach & Lee, 2018). LINC00844 is a 659 bp lncRNA transcribed from chromosome chr10q21.1 and comprises two exons (Wang et al., 2006; Kimura et al., 2006). N-myc downstream-regulated gene 1 (NDRG1) belongs to an NDR- and an α∕β hydrolase-fold region family, and is known to promote adipogenesis and sustain adipocyte function by inducing peroxisome proliferator-activated receptor gamma (PPARα) expression and C/Ebpβ activity under physiological conditions (Cai et al., 2017). In differentiated normal epithelial cells, NDRG1 retains the stability of tight junctions by regulating the expression of claudin-9 (Gao et al., 2017). However, NDRG1 expression was found to be upregulated in patients with HCC as compared with that in healthy controls and correlated with poorer outcomes (Cheng et al., 2011). NDRG1 expression may be influenced by many factors, especially hypoxia-inducible factor-1 (HIF-1) (Salnikow et al., 2002). Previous studies have shown that NDRG1 is involved in tumour invasion and metastasis (Li et al., 2019). A study reported that NDRG1 overexpression may inhibit the expression of E-cadherin and enhance the expression of Snail, and consequently regulate tumour growth and metastasis in oesophageal squamous cell carcinoma (Ai et al., 2016). Another study reported that LINC00844 exerted its antitumour activity by regulating the expression of NDRG1 in prostate cancer (Lingadalli et al., 2018). However, the biological function and underlying molecular mechanisms of LINC00844 in HCC are unclear.

Materials & Methods

Gene expression profiles

We downloaded the gene expression profiling data from The Cancer Genome Atlas database (TCGA, http://portal.gdc.cancer.gov/), which is available for free. A total of 424 tissue samples comprising 371 primary HCC samples, 3 recurrent HCC samples, and 50 normal liver tissue samples were included, and 254 clinical data were found to be complete. DESeq and edgeR based on the R language and t-test were used to evaluate differentially expressed genes.

Cell culture

The human HCC cell lines (HCCLM9, SMMC-7721, SK-Hep1, and HepG2) and a normal hepatic epithelial cell line (L02) were purchased from the Cell Bank of Type Culture Collection (CBTCC, Chinese academy of sciences, Shanghai, China), and these cell lines were cultured in Dulbecco’s modified Eagle’s medium (DMEM; Invitrogen Life Technology Inc., Carlsbad, CA) supplemented with 10% foetal bovine serum (FBS; Gibco, Grand Island, NY, USA), 100 µg/mL streptomycin, and 100 U/mL penicillin at 37 °C in a 5% CO2 incubator.

Tissue specimens

Forty HCC tissues and paired adjacent non-tumour tissues were collected from the Zhongnan Hospital of Wuhan University from April 2018 to April 2019. Surgically resected tissue samples were immediately frozen in RNAlater (Qiagen, Hilden, Germany) and subsequently stored at −80 ° C until use. The patients provided signed informed consent, and the study was approved by the Medical Ethics Committee of Zhongnan Hospital of Wuhan University. The approval number is 2019016.

Real-time quantitative polymerase chain reaction (RT-qPCR)

Total RNA was isolated from HCC tissues or cells by TRIZOL Reagent (Yeasen Biotech, Shanghai, China) following the manufacturer’s instructions. After extraction, RNA samples were reverse transcribed by a reverse transcription kit (Yeasen Biotech, Shanghai, China) and its concentration was determined using a NanoDrop ND-1000 spectrophotometer (Thermo Fisher Scientific, Wilmington, DE). The Hieff qPCR SYBR Green Master Mix kit (Yeasen Biotech, Shanghai, China) was used for real-time quantitative polymerase chain reaction (RT-PCR). Expression levels were normalised against glyceraldehyde-3-phosphate dehydrogenase (GAPDH) level, and the 2ΔΔCT method was used to calculate relative fold change. All primers were designed and synthesised by Wuhan Servicebio Technology CO., LTD (Wuhan, China) and their sequences are listed in Table 1.

Plasmid construction, lentiviral packaging, and cell transfection

Full-length LINC00844 was amplified and cloned into a lentiviral expression vector pEZ-Lv201 (Generay Biotech Co. Ltd, Shanghai, China) and subjected to lentiviral packaging (Generay Biotech Co. Ltd, Shanghai, China). Cell lines were infected with Lv-LINC00844 viruses, while Lv-NC viruses served as the negative control. Efficiency was confirmed with RT-qPCR.

Transwell assay

Transwell assay was used to detect the migration and invasion abilities of cells. A 24-well transwell chamber without or with matrigel (Corning Life Sciences, Corning, NY, USA) was used to detect cell migration and invasion rates. In brief, a 100 µL cell suspension (5 ×104 cells/well) in a serum-free medium was seeded into the upper chamber, while the lower chamber was filled with 600 µL of DEME containing 10% FBS. Cells were incubated on the membranes for 24 h, followed by fixation of the migrated cells in 4% paraformaldehyde for 20 min and staining with 0.1% crystal violet in 1× phosphate-buffered saline (PBS) for 30 min. Cells were counted from five random fields under a microscope, and the average number of cells per field was determined. All experiments were performed in triplicates.

Table 1 Primer sequences for Reverse Transcription-Quantitative Polymerase Chain Reaction.

LINC00844, Long intergenic non-protein coding RNA 844; NDRG1, N-myc downstream regulated 1; GAPDH, glyceraldehyde-3-phosphate dehydrogenase.

Gene	Primer sequence (5′–3′)	
LINC00844	CTTGATGCAGTCTGATAGGAGGAT	
TCTGCCATACTGTTTCTGGTTCA	
NDRG1	TACCGCCAGCACATTGTGAA	
GCCACAGTCCGCCATCTT	
GAPDH	GGAAGCTTGTCATCAATGGAAATC	
TGATGACCCTTTTGGCTCCC	

Cell proliferation assay

Cell counting kit-8 (CCK-8, Yeasen Biotech, Shanghai, China) was used to detect cell proliferation. Cells were plated in 96-well plates in triplicates at 2 ×103 cells/well and incubated in a 5% CO2 atmosphere at 37 ° C. The absorbance of each well at 450 nm wavelength was measured every 24 h as per the instructions of the manufacturer for 4 consecutive days obtain a growth curve. All experiments were performed in triplicates.

Immunohistochemistry (IHC)

Tumour tissues were fixed in 4% paraformaldehyde, dehydrated, embedded in paraffin, and cut into 4-µm-thick sections. Briefly, sections were dewaxed in xylene and rehydrated through graded alcohol solutions. Endogenous peroxidase activity was quenched with hydrogen peroxide and renovated antigen. The slides were incubated overnight with 10% serum to block any nonspecific binding sites. Slides were then treated with anti-NDRG1 antibody (1:400 dilution, Wuhan Proteintech Group, China) at 4 °C. The sections were eventually incubated with a peroxidase-conjugated polymer for 30 min, and a 3,3′-diaminobenzidine (DAB) system (Beyotime Institute of Biotechnology, Jiangsu, China) was used for detection.

Western blot analysis

Total protein was extracted from HepG2 and HCCLM9 cells using radioimmunoprecipitation assay (RIPA) lysis buffer (Wuhan Servicebio Biotechnology, China), and the protein concentration was measured with a bicinchoninic acid (BCA) assay kit (Wuhan Google Biotechnology Co., Ltd). Equal amounts of protein lysates were separated by sodium dodecyl sulphate polyacrylamide gel electrophoresis (SDS-PAGE; Wuhan Google Biotechnology Co., Ltd, China) and the separated bands were transferred onto nitrocellulose membranes. The membranes were incubated overnight at 4 ° C with a primary rabbit anti-NDRG1 antibody (1:800, Wuhan Proteintech Group, China). Proteins were detected using an enhanced chemiluminescence (ECL) method as previously described. GAPDH was used as a loading control.

Statistical analysis

All data are presented as means ± standard error of the mean (SEM). The Kaplan–Meier method was used for survival analysis. Correlation between LINC00844 expression and clinicopathological variables was analysed with the Student’s t-test or one-way analysis of variance (analysis of variance [ANOVA], t-test for two-group comparisons or one-way ANOVA for other cases). Statistical analysis was performed using GraphPad Prism 7 and SPSS 18.0. P <0 .05 was considered statistically significant.

Results

Expression of LINC00844 is significantly downregulated in HCC tissues and cells

The top five upregulated lncRNAs and top five downregulated lncRNAs (adjusted P value <0.05 and |logFC| ≥ 2) are shown in the heat map, and included LINC00844 (Ensemble ID:ENSG00000237949, Fig. 1A and Table 2). To understand the expression pattern of LINC00844 in different tumours, we extracted LINC00844 from 31 kinds of tumours from Gene Expression Profiling Interactive Analysis (GEPIA) (http://gepia.cancer-pku.cn/) and found that LINC00844 expression was low in most tumours, including HCC (Fig. 1B). LINC00844 was downregulated in liver HCC (LIHC) from GEPIA (Fig. 2A). In our study, the relative expression level of LINC00844 was detected in HCC tissues (n = 40) and their adjacent non-tumour tissues (n = 40) with RT-qPCR. The relative expression level of LINC00844 was significantly downregulated in HCC tissues as compared with that in paired adjacent non-tumour tissues. LINC00844 expression downregulation (> 2 fold change) was detected in 87.5% (35/40) of HCC tissues (Figs. 2F and 2G, P < 0.01). The relative expression level of LINC00844 correlated with histological grade from GEPIA (Fig. 2B, P = 0.0377). However, GEPIA showed that patients with low LINC00844 expression had no obvious association with overall survival and disease-free survival (Figs. 2C and 2D). Moreover, the area under curve (AUC) of LINC00844 expression was 0.854 (Fig. 2E, 95% confidence interval [CI]: 0.805–0.886, P < 0.05). Thus, LINC00844 has a moderate diagnostic value in HCC.

Figure 1 Analysis of the expression of LINC00844 in TCGA and GEPIA databases.

(A) Heat map of the top five upregulated and top five downregulated lncRNAs. (B) Overview of LINC00844 expression in multiple tumour entities with a notable decrease in tumours as compared with that in normal tissues. The image is derived from GEPIA.

Correlation between LINC00844 expression downregulation and clinicopathological features of patients with HCC

To evaluate the clinical significance of LINC00844 in HCC, we explored the correlation between LINC00844 expression and the clinicopathological features of patients with HCC. The results are shown in Table 3. A statistically significant correlation was detected between LINC00844 expression level and sex (Fig. 3A, Male versus Female, P < 0.01), histological grade (Fig. 3C, G1 + G2 versus G3 + G4, P < 0.01), advanced American Joint Committee on Cancer (AJCC) stage (Fig. 3E, I + II versus III + IV, P < 0.05), and vascular invasion (Fig. 3F, None versus Micro versus Macro, P < 0.01). However, no significant association was observed between LINC00844 expression and ethnicity (Fig. 3B, Asian versus White, P > 0.05) and T classification of TNM stage (Fig. 3D, T1 + T2 versus T3 + T4, P > 0.05). The results are based on the original data from TCGA. Meanwhile, To evaluate the clinical roles of LINC00844 in HCC, we also explored the correlation between LINC00844 expression and the clinicopathological factors with 40 HCC patients , and results was shown in Table 4. Lower expression of LINC00844 was closely correlated with multiple clinical pathological features including pathological stage (P = 0.0484), portal vein tumor thrombus (P = 0.0471) and TNM (P = 0.0187).

Table 2 The top 5 upregulated lncRNAs and the top five downregulated lncRNA.

Regulation	Ensemble ID	Gene symbol	Log2FC	P-value	FDR	
Up-regulation	ENSG00000272405	AL365181.3	3.50	4.58E−08	1.71E−07	
ENSG00000230733	AC092171.2	2.59	2.81E−23	9.38E−22	
ENSG00000245694	CRNDE	2.48	9.23E−10	4.43E−09	
ENSG00000265688	MAFG-AS1	2.23	9.61E−15	9.43E−14	
ENSG00000262877	AC110285.2	2.07	1.30E−13	1.09E−12	
Down-regulation	ENSG00000205866	FAM99A	−4.61	3.37E−39	5.89E−37	
ENSG00000237949	LINC00844	−4.27	1.73E−32	1.89E−30	
ENSG00000250056	LINC01018	−4.12	6.63E−18	1.00E−16	
ENSG00000248740	LINC02428	−3.75	1.14E−33	1.41E−31	
ENSG00000248709	AC008549.1	−3.59	4.77E−31	4.01E−29	

Figure 2 LINC00844 is significantly downregulated in HCC.

(A) Expression of LINC00844 was analysed in HCC from GEPIA. *P < 0.01. (B) The relationship between the expression of LINC00844 and HCC pathological stage from GEPIA. *P < 0.05. (C and D) Kaplan-Meier analysis of the correlation between LINC00844 expression and disease-free survival and overall survival in GEPIA. P > 0.05. (E) ROC curve of LINC00844 in HCC derived from the original data from TCGA database. P < 0.01, AUC = 0.854. (F and G) Expression and fold change of LINC00844 in 40 pairs of HCC tissues and adjacent non-tumour tissues. Downregulation of LINC00844 (> two-fold change) expression was detected in 87.5% (35/40) of HCC. **P < 0.01. GAPDH was used as control. All results are obtained from at least three independent experiments.

LINC00844 expression upregulation inhibits HepG2 and HCCLM9 cell proliferation, migration, and invasion

We observed that HCC cell lines (HepG2, HCCLM9, SMMC-7721, and SK-Hep1) expressed significantly lower levels of LINC00844 than the normal hepatic epithelial cell line L02 (Fig. 4A, P < 0.01). LINC00844 expression was relatively lower in HepG2 and HCCLM9 cell lines than in SK-Hep1 and SMMC-7721 cell lines. Therefore, we chose HepG2 and HCCLM9 cell lines for the following gain-of-function studies. Relative LINC00844 expression in HepG2 and HCCLM9 cell lines was significantly increased upon infection with Lv-LINC00844; Lv-NC viruses acted as a negative control (Fig. 4B, P < 0.01). As per the results of the CCK-8 assay, the OD450 values of HepG2 and HCCLM9 cell lines from Lv-LINC00844 group were remarkably lower than those for the cells from Lv-NC group at 0, 24, 48, and 72 h (Figs. 4C and 4D, P < 0.01). Cell migration and invasion abilities were analysed with transwell assays and found to be significantly decreased for HepG2 and HCCLM9 cell lines from Lv-LINC00844 group as compared with those from Lv-NC group (Figs. 4E, 4F, 4G and 4H, P < 0.01).

Table 3 The correlation between LINC00844 expression and the clinicopathological features of patients with HCC.

Clinicopathological parameters	LINC00844	
	N	Mean+SEM	T	P-value	
Tissues					
Normal liver	50	4.953 ± 0.154	6.437	<0.000	
HCC	345	1.803 ± 0.185	
Sex					
Male	167	2.411 ± 0.271	4.710	<0.000	
Female	87	0.288 ± 0.344	
Race					
Asian	114	1.837 ± 0.332	0.620	0.536	
White	140	1.559 ± 0.301	
T (tumor)					
T1+T2	205	1.846 ± 0.242	1.493	0.137	
T3+T4	49	1.005 ± 0.546	
Stage					
I+II	202	1.196 ± 0.243	2.066	0.040	
III+IV	52	0.783 ± 0.530	
Histological grade					
G1+G2	152	2.305 ± 0.279	3.481	0.001	
G3+G4	102	0.758 ± 0.350	
Vascular invasion					
None	168	2.050 ± 0.264	F = 5.215	0.006	
Micro	71	1.352 ± 0.430	
Macro	15	−0.856 ± 0.954	

NDRG1 is overexpressed in clinical HCC tissues and cell lines and negatively correlates with LINC00844

To investigate the mechanism of action of LINC00844 in HCC tumourigenesis, we analysed the mRNA level of NDRG1 in HCC tissues and cells and investigated the relationship between NDRG1 and LINC00844 expression. NDRG1 expression was upregulated in LIHC from GEPIA (Fig. 5A). We also examined the expression level of NDRG1 mRNA with RT-qPCR and found that NDRG1 level was markedly elevated in 20 HCC tissues as compared with that in paired adjacent non-tumour tissues (Fig. 5D, P < 0.01). The expression level of NDRG1 correlated with histological grade from GEPIA (Fig. 5B, P = 0.0126). GEPIA demonstrated that high NDRG1 expression level showed a significant association with overall survival in patients with HCC (Fig. 5C). The expression level of LINC00844 was negatively associated with that of NDRG1 in 20 paired clinical HCC tissues (Fig. 5E, P < 0.05, R2 = 0.3043, Pearson’s correlation). IHC assay was performed to investigate the expression of NDRG1 protein in HCC tissues (Fig. 5F). An increase in LINC00844 expression resulted in a decrease in the mRNA and protein levels of NDRG1 in HepG2 and HCCLM9 cells (Fig. 5G).

Figure 3 Correlation between LINC00844 downregulation and clinicopathologic features in 254 HCC samples and 50 normal liver tissues based on the original data from TCGA.

(A, C, E, and F) The relationship between LINC00844 expression and sex (Male versus Female, P < 0.001), histological grade (G1 + G2 versus G3 + G4, P < 0.001), advanced American Joint Committee on Cancer (AJCC) stage (I + II versus III + IV, P = 0.0399), and vascular invasion (None versus Micro versus Macro, P = 0.006). (B and D) The relationship between the expression of LINC00844 and ethnicity (Asian versus White, P = 0.5361) and T classification of TNM stage (T1 + T2 versus T3 + T4, P = 0.1366).

Table 4 The relationships of LINC00844 expression with clinicopathological parameters in HCC based on 40 HCC patients.

Variable	Number of cases	LINC01554 expression	P-value	
		Low (N = 20)	High (N = 20)		
Gender				0.0915	
Male	33	14	19	
Female	7	6	1	
Age(year)				0.6948	
≤50	8	3	5	
>50	32	17	15	
HBV infection				0.2049	
Negative	19	7	12	
Positive	21	13	8	
Serum AFP(ng/mL)				0.4506	
≤8.78	9	6	3	
>8.78	31	14	17	
Pathological stage				0.0484	
I+II	25	9	16	
III+IV	15	11	4	
Cirrhosis				0.4801	
No	11	7	4	
Yes	29	13	16	
PVTT				0.0471	
No	35	15	20	
Yes	5	5	0	
TNM(AJCC)				0.0187	
I + II	26	9	17	
III+IV	14	11	3	
Notes.

PVTT portal vein tumor thrombus

AJCC American Joint Committee on Cancer

* P < 0.05.

Discussion

Genome-wide sequencing has revealed the involvement of more and more lncRNAs in different types of cancers (Bach & Lee, 2018). Lingadahalli et al (Lingadalli et al., 2018) performed in vivo and in vitro experiments to confirm the participation of LINC00844 in tumour metastasis partly through its effect on the expression of NDRG1 in prostate cancer. High expression of lncPARP1 may serve as an adverse prognosis factor for HCC (Qi et al., 2018). In our study, we used TCGA and GEPIA and found that the expression level of LINC00844 was lower in HCC tissues than in normal samples. We also verified the dramatic decrease in LINC00844 expression in 40 paired primary HCC tissues and cell lines as compared with that in the paired adjacent non-tumour tissues and normal liver cell lines. We obtained data on LINC00844 expression and relative clinical features from TCGA and found that low LINC00844 expression was highly associated with late clinical stage, poor histological grade, and vascular invasion in patients with HCC. And we discovered lower expression of LINC00844 was positively correlated with pathological stage, portal vein tumor thrombus and TNM in 40 HCC patients. These findings were the approximate homology with TCGA dataset. These results suggest LINC00844 may have a clinical significance in the diagnosis of HCC. To investigate the biological role of LINC00844 in HCC, LINC00844 expression was induced in HCC cells (HepG2 and HCCLM9). As a result, LINC00844 overexpression was shown to significantly inhibit proliferation and suppress migration and invasion of HepG2 and HCCLM9 cell lines. Based on these results, we suggest that LINC00844 may act as a tumour suppressor in HCC.

Figure 4 Upregulation in LINC00844 expression represses the proliferation, migration, and invasion of HepG2 and HCCLM9 cells.

(A) The relative expression of LINC00844 in HCC cell lines (HepG2, SK-Hep1, HCCLM9, and SMMC-7721) and a human normal liver cell line (L02). **P < 0.01. GAPDH was used as control. (B) LINC00844 expression in HepG2 and HCCLM9 cells after transfection with Lv-LINC00844 was detected with RT-qPCR. **P < 0.01. GAPDH was used as control. (C, D) Proliferation was detected with the CCK-8 assay in HepG2 and HCCLM9 cells, **P < 0.01. (E, F, G and H). The migration and invasion of HepG2 and HCCLM9 cells were examined with transwell migration and invasion assays. **P < 0.01. All experiments were performed in triplicates.

As a key regulator, lncRNAs are involved in the regulation of gene expression in cancers, and play an important role in chromatin remodelling complexes and gene regulation at transcriptional and transcriptional levels (Schmitt & Chang, 2016; Quinn & Chang, 2016). For instance, the lncRNA PCAT6 was shown to inhibit non-small cell lung cancer cell proliferation, apoptosis, and metastasis through epigenetic mechanisms (Shi et al., 2018). MIR31HG suppressed HCC proliferation and metastasis by serving as an miR-575 sponge in vitro and in vivo (Yan et al., 2018). The lncRNA Xist is known to increase histone trimethylation level and reduce the activity of X chromosome (Simon et al., 2013). A recent report showed that LINC00844 inhibited tumour metastasis by regulating the expression of NDRG1 in prostate cancer (Lingadalli et al., 2018). Growing evidence has highlighted the regulatory role of NDRG1 in a variety of functions, including cellular differentiation, activation of p53, and cellular apoptosis and metastasis (Sibold et al., 2007; Byun et al., 2018; Chen et al., 2015). Yan et al. (Yan et al., 2008)found that NDRG1 expression is generally upregulated in HCC tissues as compared with that in normal samples, particularly in recurrent and metastatic HCC. NDRG1 overexpression enhances aerobic glycolysis and imparts growth advantages to cells, induces the expression of hypoxia-associated genes, and causes lipid metabolism dysfunction in breast cancer (Sevinsky et al., 2018). A report revealed the role of the lncRNA CCAT2 in promoting cellular proliferation and metastasis through the upregulation of NDRG1 expression in HCC (Liu et al., 2019). Consistent with these results, we confirm that NDRG1 expression was upregulated in 20 pairs of HCC tissues and that patients with higher NDRG1 expression showed poorer clinical prognosis and histological grade. The expression of NDRG1 mRNA was negatively correlated with LINC00844 in HCC tissues, and LINC00844 overexpression could partly inhibit the expression of NDRG1 in HepG2 and HCCLM9 cells. How LINC00844 controls tumour migration and invasion in HCC warrants further studies.

Figure 5 NDRG1 was highly expressed in clinical HCC tissues and cell lines and negatively correlated with LINC00844.

(A) Expression of NDRG1 was analysed in HCC from GEPIA. *P < 0.01. (B) The relationship between NDRG1 expression and pathological stage. (C) Kaplan–Meier analysis to detect the correlation between NDRG1 expression and overall survival for HCC samples from GEPIA. *P < 0.05. (D) qRT-PCR revealed the relative expression level of NDRG1 in 20 pairs of HCC tissues and adjacent non-tumour tissues. P < 0.05. GAPDH was used as control. (E) The correlation between LINC00844 expression level and NDRG1 mRNA level in 20 HCC tissues. The Δ CT values were subjected to Pearson’s correlation analysis. ΔCT value was determined by subtracting GAPDH Ct value from LINC00844 ΔCt value and NDRG1 Ct value. P < 0.05, R2 = 0.3043. GAPDH was used as control. (F) Representative immunostaining to detect NDRG1 protein expression in HCC tissues and paired adjacent non-tumour tissues. (G) The changes in NDRG1 mRNA and protein levels were examined in HepG2 and HCCLM9 cells overexpressing LINC00844. **P < 0.01. GAPDH was used as control. All results are obtained from at least three independent experiments.

Conclusions

In summary, we demonstrated that the expression of LINC00844 is downregulated in HCC samples and cell lines. LINC00844 overexpression inhibits HCC cell proliferation and invasion, at least in part, through the suppression of NDRG1 expression. Thus, LINC00844 may be potentially used as a promising prognostic biomarker and therapeutic target for HCC.

Supplemental Information

Figure S1 “Regression analysis of LINC00844 and NDRG1 expression in I + II pathological stages”

Click here for additional data file.

Figure S2 “Regression analysis of LINC00844 and NDRG1 expression in III + IV pathological stages”

Click here for additional data file.

File S1 The raw data of expression of 10 LncRNAs in 371 primary hepatocellular carcinoma and 50 normal liver tissue samples from TCGA dataset

Click here for additional data file.

File S2 The raw data of ROC from TCGA dataset

Click here for additional data file.

File S3 Raw data of expression of LINC00844 in 40 pairs of HCC tissues and 40 paired adjacent non-tumor tissues detected by RT-qPCR assay

Click here for additional data file.

File S4 Raw data of fold change of LINC00844 in 40 pairs of HCC tissues compared with 40 adjacent non-tumor tissues by RT-qPCR assay

Click here for additional data file.

File S5 Raw data of expression of LINC00844 in HCC lines by RT-qPCR assay

Click here for additional data file.

File S6 Raw data of LINC00844 expression in HepG2 and HCCLM9 cells after transfection of Lv-LINC00844 was detected by qRT-PCR

Click here for additional data file.

File S7 The raw data in 254 expression of LINC00844 and clinical data from TCGA dataset

Click here for additional data file.

File S8 The original CCK8 data of in HepG2 and HCCLM9 cells after transfection of Lv-LINC00844

Click here for additional data file.

File S9 The original migration assay data of in HepG2 and HCCLM9 cells after transfection of Lv-LINC00844

Click here for additional data file.

File S10 The original invasion assay data of in HepG2 and HCCLM9 cells after transfection of Lv-LINC00844

Click here for additional data file.

File S11 Raw data of expression of NDRG1 in 20 pairs of HCC tissues and 20 paired adjacent non-tumor tissues detected by RT-qPCR assay

Click here for additional data file.

File S12 Raw data of correlation between expression of LINC00844 and expression of NDRG1 mRNA in 20 HCC tissues

Click here for additional data file.

File S13 The original image of immunostaining staining of NDRG1 protein of 3 HCC tissues and 3 paired adjacent non-tumor tissues

Click here for additional data file.

File S14 Raw data of NDRG1 expression in HepG2 and HCCLM9 cells after transfection of Lv-LINC00844 was detected by qRT-PCR

Click here for additional data file.

File S15 The original Western Blot for the band of NDRG1 and GAPDH

Click here for additional data file.

File S16 All the differentially regulated lncRNAs and the order of their ranking

Click here for additional data file.

We would like to thank all tutors and friends.

Abbreviations

lncRNA long noncoding RNA

LINC00844 long intergenic non-protein coding RNA 844

HCC hepatocellular carcinoma

NDRG1 N-myc downstream-regulated gene 1

RT-qPCR real-time quantitative polymerase chain reaction

TCGA The Cancer Genome Atlas

CCK-8 cell counting kit-8

AJCC American Joint Committee on Cancer

HBV hepatitis B virus

HCV hepatitis C virus

HIF-1 hypoxia inducible factor-1

DMEM Dulbecco’s modified Eagle’s medium

GAPDH glyceraldehyde-3-phosphate dehydrogenase

PBS phosphate-buffered saline

IHC immunohistochemistry

DAB 3,3′-diaminobenzidine

RIPA radioimmunoprecipitation assay

SDS sodium dodecyl sulphate

PAGE polyacrylamide gel electrophoresis

LIHC liver hepatocellular carcinoma

GEPIA gene expression profiling interactive analysis

AUC area under curve

OD450 optical density 405 nm

Additional Information and Declarations

Competing Interests

Author Contributions

Human Ethics

Data Availability

The authors declare there are no competing interests.

Wei Zhou, Kang Huang, Ling Li and Qifa Ye conceived and designed the experiments, performed the experiments, analyzed the data, prepared figures and/or tables, authored or reviewed drafts of the paper, and approved the final draft.

Qiuyan Zhang performed the experiments, analyzed the data, prepared figures and/or tables, authored or reviewed drafts of the paper, and approved the final draft.

Shaojun Ye, Zibiao Zhong, Cheng Zeng and Guizhu Peng performed the experiments, analyzed the data, authored or reviewed drafts of the paper, and approved the final draft.

The following information was supplied relating to ethical approvals (i.e., approving body and any reference numbers):

The Medical Ethics Committee of Zhongnan Hospital of Wuhan University approved this research (2019016).

The following information was supplied regarding data availability:

The raw measurements are available in the Supplemental Files.

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
