# Peer review of "LINC00844 promotes proliferation and migration of hepatocellular carcinoma by regulating NDRG1 expression"

_PeerJ, doi:10.7717/peerj.8394_

## Round 0.1 · original submission · Minor Revisions

I am pleased to underline the quality of the research proposed in the manuscript. Nevertheless, following the comments of the reviewers, I would dedicate more attention to improving the use of language, spelling mistakes and of course, addressing the suggestions pointed out by the reviewers.

Reviewer 1 ·

Basic reporting

1. There is a need to improve English. Many grammatical and spelling mistakes can be found throughout the manuscript.
2. Authors should provide an index to all the abbreviations used or expand them accordingly.
4. In the introduction, line 55, the information about the LINC00844 is insufficient. Authors should expand on this and include relevant information to emphasize as to why studying LINC00844 is important.
5. In lines 61-62, authors should cite the appropriate manuscript to support their statement and the current citation(#16) does not support it.
6. Typological Error in abstract (LINC008444).
7. Line 151, 'markedly' should be changed to 'significantly' as the difference in LINC00844 expression is statistically significant.
8. Fig2A and 5A, please mention the p-value in the figure.
9. In line 163, what is LINC01554 ? hope it is a typo.

Experimental design

The primary research aim is within the Aims and scope of the journal.
The research questions are well defined. The experiments are performed with appropriate controls. The experiments performed in tissue specimens were collected in their institute with patient consent and ethical approval.

Validity of the findings

Authors present an interesting story of how LINC00844 regulates hepatocellular carcinoma cell migration by regulating the expression of NDRG1 and the results are novel to the field of HCC.

The article can be improved if the authors can address the following points.
1. Authors describe the methodology to identify differentially regulate lncRNAs but do not show the results for this analysis. In Figure 1A, they only show the expression of top 5 lncRNAs. Authors must include a figure/supplementary table of all the differentially regulated lncRNAs and the order of their ranking.
2. Figures 2B and 5B, regression analysis of LINC00844 and NDRG1 expression in different pathological stages should be performed and presented.
3. Authors should provide sufficient evidence to support to rational of exploring the relationship of LINC00844 expression with various clinical and pathological features described in Fig 3. Authors should also explain the biological relevance of all the differences in LINC00844 expression observed.
4. Fig 2H and I should be included in Fig 4. It is unclear to me as to why they have been placed there.

Reviewer 2 ·

Basic reporting

1. The authors need to rewrite or extend some parts as it is the case of the introduction. We suggest some additional references that might help to provide a better overview of the field and highlight the pivotal role of NDRG1 in non-pathological and pathological conditions.

2. We strongly advise the authors to consider a proof-reading service which will substantially improve the overall quality of the manuscript.

References
1. Luo Q et al. FOXQ1/NDRG1 axis exacerbates hepatocellular carcinoma initiation via enhancing crosstalk between fibroblasts and tumor cells. Cancer Lett. 2018 Mar 28;417:21-34
2. Cai K et al.Ndrg1 promotes adipocyte differentiation and sustains their function. Sci Rep. 2017 Aug 3;7(1):7191
3. Lu WJ, NDRG1 promotes growth of hepatocellular carcinoma cells by directly interacting with GSK-3β and Nur77 to prevent β-catenin degradation. Oncotarget. 2015 Oct 6;6(30):29847-59
4. Jung EU et al.Hypoxia and retinoic acid-inducible NDRG1 expression is responsible for doxorubicin and retinoic acid resistance in hepatocellular carcinoma cells. Cancer Lett. 2010 Dec 1;298(1):9-15
5. Akiba et al. N-myc downstream regulated gene 1 (NDRG1)/Cap43 enhances portal vein invasion and intrahepatic metastasis in human hepatocellular carcinoma. Oncol Rep. 2008 Dec;20(6):1329-35.

Experimental design

no comment

Validity of the findings

no comment

Additional comments

The authors reported a novel and important role of LINC00844 in the onset of liver cancer. The authors addressed the hypothesis that LINC00844 might affect the levels of NDRG1 from different approaches by providing in vitro data as well as interesting data from human samples. The study is conceptually well designed and it would be suitable for publication in PeerJ, However, there are just a few minor points that need to be addressed.

---

## Round 0.2 · accepted · Accept

The reviewers agreed that this version is improved. I would like to congratulate the authors.

Please, check the comments of the reviewers anyway.

Reviewer 1 ·

Basic reporting

The authors have addressed all the comments and have made satisfactory changes. The manuscript is now written in professional English. It is my opinion that the manuscript is now acceptable for the broad readership of Peer J.

Experimental design

No changes required.

Validity of the findings

have addressed all the comments. No further revision required.

Reviewer 2 ·

Basic reporting

no comment

Experimental design

no comment

Validity of the findings

no comment

Additional comments

The authors have extended the introduction part and have made a clear improvement in the quality of the manuscript.
However, in the methods part some headlines are wrong, the protocol for plasmids and lentiviral packaging and cell transfection contains a protocol for RNA isolation so it is inverted and it must be change. In addition, there are still a few grammar mistakes, thus, we strongly advise the authors to make a second proofreading to make sure the spelling and the use of the verb tenses are correct.